# Determinants of change in long-acting or permanent contraceptives use in Ethiopia; A multivariate decomposition analysis of data from the Ethiopian demographic and health survey

**Gedefaw Abeje Fekadu** [1,2]*, **Akinyinka O. Omigbodun**[3], **Olumuyiwa A. Roberts**[3], **Alemayehu Worku Yalew**[4]

**1** Institute of Life and Earth Sciences (including health and Agriculture) Pan African University, University of Ibadan, Ibadan, Nigeria, **2** College of Medicine and Health Sciences, School of Public Health, Bahir Dar University, Bahir Dar, Ethiopia, **3** Department of Obstetrics and Gynecology, College of Medicine, University of Ibadan, University College Hospital, Ibadan, Nigeria, **4** School of Public health, College of Health Sciences, Addis Ababa University, Addis Ababa, Ethiopia

\* abejegedefaw@gmail.com

**Data Availability Statement:** For this analysis, we used the 2005 and 2016 Ethiopian demographic and health survey data sets. The data was

# Abstract

## Background

There has been an increase in the uptake of long-acting or permanent contraceptive methods (LAPMs) in Ethiopia. Identifying the factors associated with this change is important for designing interventions that will further accelerate the uptake. This study was done to identify components of, and factors associated with, changes in the use of LAPMs in Ethiopia.

## Methods

Information about 16,336 married or in-union reproductive-age women were extracted from the 2005 and 2016 Ethiopian Demographic and Health Surveys (EDHS). Normalized weighting was used to compensate for disproportionate sampling and non-response in the survey. The two data sets were merged and analyzed using multivariate decomposition analysis.

## Result

From 2005 to 2016, the use of LAPMs increased by 12.0 percentage points. Changes in the characteristics of women (compositional factors) were responsible for nearly 7.0% of the observed difference. Most of the change (92.0%) was attributable to differences in the effects of characteristics. Age, working status, woman's occupation, concordance on the desired number of children between women and their partners, and a visit by health workers in the 12 months before the survey were all significantly associated with the change.

accessed from The DHS Program website (https://dhsprogram.com/data/available-datasets.cfm) for free. We do not have special access privileges to this data. All authors can access the data from this website. To get the data, authors should register and log in. While login, they requested to state the project title, co-researchers' name and email and a brief description of the study. After that, the researchers continue to select the country and the data set. Within a few days, he/she will get permission to download the data via email. After the permission, the researcher can log in and select the specific data with the format he/she wants.

**Funding:** The authors received no specific funding for this work.

**Competing interests:** The authors have declared that no competing interests exist.

## Conclusion

The contribution of variation in the survey population structure was not significant for the observed change. The change in the use of LAPMs was mainly due to behavioral changes among older, educated and working women, and women visited by health workers.

## Introduction

Unintended pregnancy is a major global problem causing health, social and economic challenges among sexually active women. Globally, about 44% of pregnancies were unintended [1]. A multi-country analysis of DHS data from Sub-Saharan counties identified that 29% of married women had unintended pregnancy ranging from 10.8% in Nigeria to 54. 5% in Namibia [2]. The 2016 Ethiopian demographic and health survey (EDHS) report identified that 25% of births in Ethiopia were unintended (8% unwanted and 17% mistimed)[3]. The main reason for high level of unintended pregnancy in Africa is low contraceptive use. In addition, contraception in Africa is dominated by short-acting methods [4, 5]. The long-acting and permanent methods (LAPMs) such as intrauterine devices (IUD), implants, vasectomy, and tubal ligation, are known to be efficient and cost-effective interventions for reducing unintended pregnancies. LAPMs can be used for 3 or more years without visiting health facilities. But these methods are underutilized in developing countries [6–8].

The trend in LAPMs use is increasing slowly in sub-Saharan Africa (SSA) [4, 9–11]. Lack of knowledge among women, dependence on the provider for information and provider-bias were among the reasons cited for low utilization of these methods [9]. Contraceptive uptake increased up to nine-fold during the last three decades in Ethiopia [4, 12]. However, similar to other African countries, it is dominated by short-acting methods [1, 3, 13]. The 2016 EDHS report identified that only 10.0% of married or in-union reproductive-age women were using LAPMs [3]. Majority of these women were using implants [3]. On the other hand, studies in various parts of the country indicated that there is high demand for LAPMs [14–16].

The Ethiopian government targeted contraceptive prevalence (CPR) of 55% at the end of 2020. Long acting or permanent methods are expected to constitute 50% of the CPR [17]. The government and many other non-governmental organizations are implementing different interventions to increase LAPMs uptake. Reports indicate that most health facilities have the basic infrastructure to provide LAPMs. All contraceptives, including LAPMs are provided free [17]. Yet, LAPMs uptake is lagging behind the national target. Identifying the factors or drivers of the change is important to formulate appropriate policies and strategies important to accelerate the uptake. But there are no studies to identify whether changes in LAPMs use are due to change in population structure or due to public health interventions. Therefore, this analysis was done to identify factors associated with change in long acting or permanent contraceptives method use.

## Methods

### Data source

We analyzed the 2005 and 2016 EDHS data collected by the Central Statistics Agency (CSA). The surveys used a list of enumeration areas, used for the 2007 housing and population census, as the sampling frame. The surveys were designed to provide key indicators at the national and regional levels. The surveys used a two-stage stratified sampling technique. In both surveys,

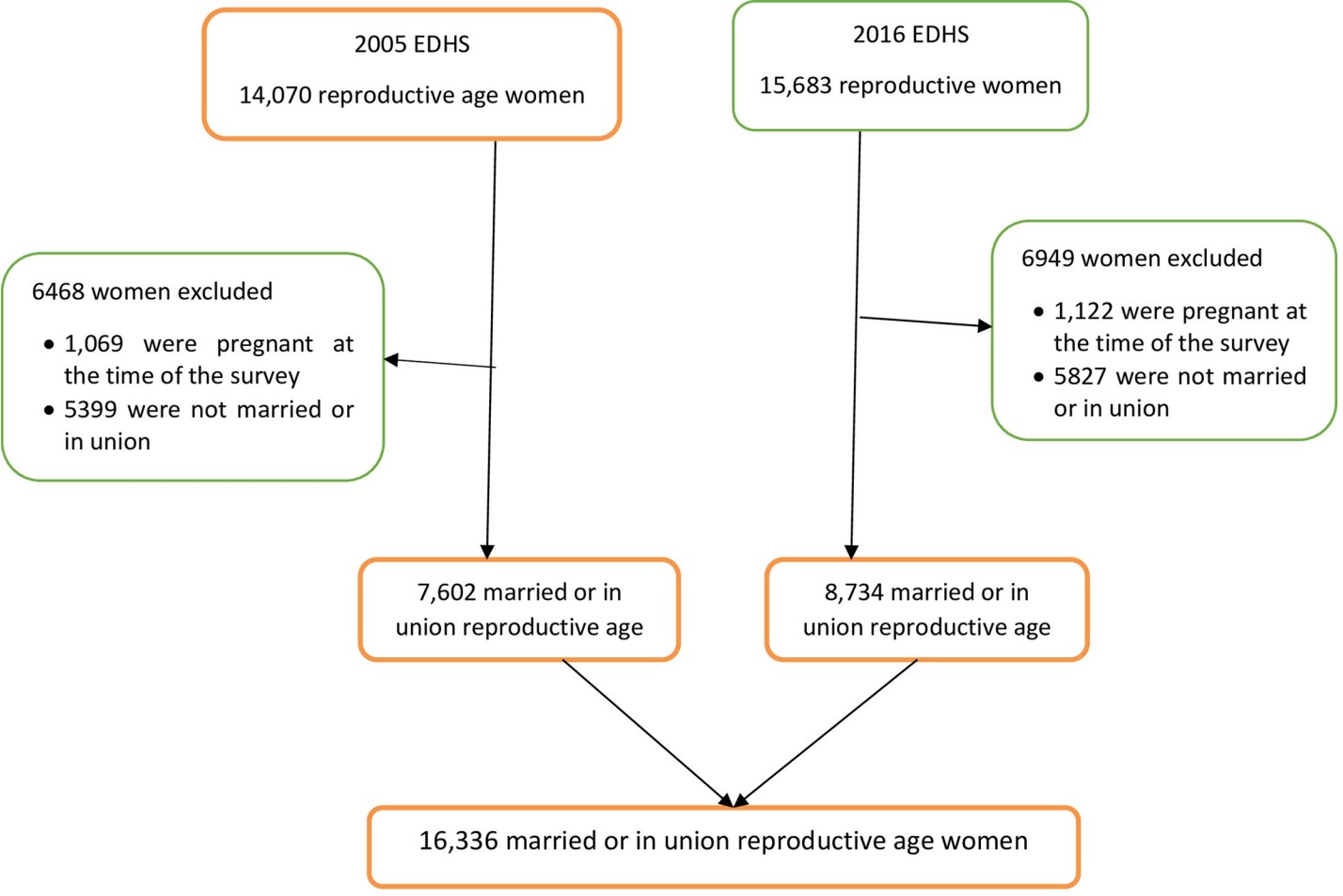

**Fig 1. Schematic presentation of the selection of women included in the analysis to identify factors associated with change in LAPM contraceptive methods use in Ethiopia.**

women aged 15–49 who were either permanent residents of the selected households or visitors who stayed in the household the night before the survey were interviewed. An interviewer-administered questionnaire was used to collect data. The details of the methodology; sampling technique, data collection, and data quality assurance are available from EDHS reports [3, 18].

A total of 29,753 reproductive-age women (women aged 15–49 years) were included in the two EDHS (14,070 in 2005 and 15,683 in 2016). From these, 18,468 (8,644 in the 2005 EDHS and 9,824 in the 2016 EDHS) were married or in-union. Women who were pregnant at the time of the survey (2,191), single, divorced and widowed women were excluded from the analysis. The reason for this is that information about contraceptive use among unmarried (never married, widowed or divorced) women is less reliable due to social desirability bias. The reason for this bias is that sexual activity among these women is taboo in Ethiopia. Unmarried women in Ethiopia are also less likely to use LAPM. Finally, 16,336 married or in-union reproductive-age women were included in the final model (Fig 1).

## Measurement

**Outcome variable.** The dependent variable for this study was long-acting or permanent contraceptive method use. In both surveys, women were asked if they were using

contraceptives and they type of contraceptive. Based on that response, the outcome variable has two categories; using long-acting or permanent contraceptive methods (implant, IUD, vasectomy and tubal ligation), coded as "1" and not using long-acting or permanent methods (using short-acting methods or not using any method), coded as "0".

**Independent variables.** Socio-Demographic Characteristics: Mothers' age at the time of interview, residence, religion, educational status, mothers' working status, sex of head of the household, occupation, wealth index and frequency of reading newspapers, listening to the radio and watching television (TV).

Fertility and Decision-Making Variables: Age at first cohabitation, the ideal number of children, number of living children, knowledge of ovulatory period, desire for more children and knowledge of fertility period.

Family Planning Program Exposure: Exposure to family planning messages on mass media and decision maker to seek health care. A woman was considered as "exposed to family planning messages on mass media" if she had read family planning messages in newspapers or magazines, or heard family planning messages on radio or TV in the preceding few months.

## Data analysis

The study employed standard (two-component) multivariate decomposition analysis technique, which can be used to analyze differences between two groups or differences between two points in time [19]. Many public health studies used this technique of analysis to identify components of a change over time and identify factors associated with change [20–25]. The analysis decomposes the differences in two points of time into two components. The first component is the change due to variation in the survey population structure and the second component is the change due to change in public health and/or changes in the behavior of the survey population [26, 27]. The decomposition analysis is based on the standard procedure of decomposing differences in which the dependent variable is a function of predictors and regression coefficients, graphically represented as $Y = F(X\beta)$.

For this analysis, the 2016 EDHS data was appended to the 2005 EDHS data using the "append" command in STATA. Since the variables in the two data sets were similar, no recoding of variables was done before merging. But after the two data sets were appended, some variables were recoded to create new categories using the recode command. Normalized weight was used to correct for disproportionate sampling and non-response in the DHS. Multicollinearity was checked using the correlation coefficient. Finally, multivariate decomposition analysis was done using the "mvdcmp" command of STATA 15.1. Before the multivariable decomposition analysis, chi-square test was done to check the presence of statistically significant difference in the distribution of women in the two surveys.

**Ethical considerations.** The 2005 and 2016 EDHS protocols were reviewed and approved by the National Ethics Review Committee of the Federal Democratic Republic of Ethiopia, Ministry of Science and Technology and the Institutional Review Board of ICF International. The data sets in STATA format were downloaded from the DHS Program after obtaining permission.

## Results

### Characteristics of married or in-union women

In both surveys, the majority of the respondents (88.6% in 2005 and 83.3% in 2016) were rural residents. Most women were not working at the time of the survey and had no formal education. There were significant differences in age at first marriage, ideal number of children and concordance on number of children among women in the two surveys (Table 1).

**Table 1. Percentage distribution of selected characteristics of married or in-union reproductive-age women in Ethiopia from the 2005 and 2016 Ethiopian Demographic and Health Surveys.**

| Characteristics of women | Percent of women in | |
|---|---|---|
| | **2005** | **2016** |
| Age*<br>• 15–24<br>• 25–34<br>• 35–49 | 24.038.937.0 | 21.142.836.1 |
| Place of residence***<br>• Urban<br>• Rural | 11.488.6 | 16.583.5 |
| Religion<br>• Orthodox<br>• Muslim<br>• Other | 46.331.821.9 | 41.633.624.8 |
| Education status***<br>• No formal education<br>• Primary<br>• Secondary or higher | 78.115.26.7 | 62.127.610.3 |
| Working status**<br>• Working<br>• Not working | 74.225.8 | 68.531.5 |
| Age at first cohabitation***<br>• Less than 20<br>• 20 or more | 84.615.4 | 79.120.9 |
| Ideal number of children**<br>No child<br>1–5<br>6 or more | 11.8<br>48.9<br>39.3 | 9.0<br>54.3<br>36.7 |
| Concordance on number of children***<br>• Both wants the same<br>• Husband wants more<br>• Husband wants fewer<br>• Do not know | 32.717.14.74.5 | 39.225.67.228.0 |
| Visit by health worker<br>• No<br>• Yes | 91.88.2 | 70.529.5 |

Note: Chi-square significant at

*Significant at 0.05

**Significant at 0.01

***Significant at<0.001

### Long-acting or permanent contraceptive methods use by selected characteristics

The use of long-acting and permanent contraceptive methods varied by socio-demographic characteristics of married or in-union reproductive age women in the two surveys. The change in long-acting and permanent contraceptive method use was positive in all categories of women. The use of long-acting and permanent contraceptive methods varied according to the fertility-related characteristics. Generally, a higher proportion of women in 2016 used long-acting and permanent contraceptive methods compared to 2005 (Table 2).

### Decomposition analysis

Overall, the use of long acting and permanent contraceptive methods changed significantly by 12.0 percentage points from 2005 to 2016. About 7.0% of the observed difference could be ascribed to the characteristics of the women in the two surveys (compositional factors).

**Table 2. Long-acting and permanent contraceptive methods use among married or in union reproductive-age women in Ethiopia and percent change by selected characteristics, 2005 and 2016 Ethiopian Demographic and Health Survey.**

| Characteristics of women | Percent of women using LAPMs in | | Percent change |
|---|---|---|---|
| | 2005 (n = 7,918) | 2016 (n = 9,127) | |
| Age<br>• 15–24<br>• 25–34<br>• 35–49 | 0.30.31.2 | 10.613.89.6 | 10.313.58.4 |
| Place of residence<br>• Urban<br>• Rural | 4.10.2 | 17.70.4 | 13.60.2 |
| Religion<br>• Orthodox<br>• Muslim<br>• Other | 1.00.30.6 | 15.35.413.8 | 14.35.113.2 |
| Education status<br>• No education<br>• Primary<br>• Secondary or higher | 0.30.93.7 | 10.910.917.2 | 10.610.013.5 |
| Working status<br>• Working<br>• Not working | 0.60.7 | 10.414.2 | 9.815.5 |
| Age at first cohabitation<br>• Less than 20<br>• 20 or more | 0.61.1 | 11.511.8 | 10.910.7 |
| Ideal number of children<br>• No child<br>• 1–5<br>• 6 or more | 0.40.90.4 | 6.315.29.0 | 5.914.38.6 |
| Concordance on number of children<br>• The same<br>• The husband wants more<br>• The husband wants less | 0.7 0.51.3 | 14.310.110.7 | 13.69.69.4 |
| Visited by health worker in the last 12 months<br>• No<br>• Yes | 0.5<br>1.7 | 10.913.3 | 10.411.6 |

After controlling for the compositional change, 93.0% of the change in LAPMs use was due to the differences in the effects of specific characteristics rather than the structural composition of the two cohorts. The effects of age, working status, concordance by husband and wife on the desired number of children, and visit by health workers in the 12 months preceding the survey were significant contributors to the change in LAPMs use. Keeping all other factors constant, about 10.0% of the increase in LAPMs use was due to behavioral change among older women (aged 35–49 years). The effect of having higher education was also significant on LAPMs use, although this effect was small. Receiving higher education was responsible for about 1.5% of the change in LAPMs use.

The effect of the difference in number of children that the woman and her husband wanted was significant for the change in LAPMs use. A greater desire for more children by the man than the woman increased LAPMs use by 1.3%. Working at the time of the survey affected LAPMs use in 2016. Compared with working mothers, non-working mothers showed a significant contribution to the observed change in LAPMs use over the decade. Visit by health worker was the other variable whose effect was responsible for the change in long-acting and permanent contraceptive methods use. The effect of visit by health workers was found associated with 2.7% change in long-acting and permanent contraceptive methods use (Table 3).

**Table 3. Decomposition of the change in long-acting and permanent contraceptive methods use among married or in union reproductive-age women in Ethiopia by selected characteristics, 2005 to 2016.**

| Characteristics of women | Percent difference in LAPMs use due to | | |
|---|---|---|---|
| | Characteristics (E) | Coefficients (C) | Total |
| Age (in years) at the time of the survey<br>• 15–24<br>• 25–34<br>• 35–24<br>• Overall | Ref.<br>-0.70.44.4 | Ref.6.8–10.0*-43.9* | Ref.6.1–9.6–39.5 |
| Age (in years) at first cohabitation<br>• <20<br>• 20 or more<br>• Overall | Ref.0.50.3 | Ref.-0.9–1.4 | Ref.-0.4–1.1 |
| Religion<br>• Orthodox<br>• Muslim<br>• Other<br>• Overall | Ref.2.60.34.3 | Ref.-9.4*7.3*13.1 | Ref. -6.87.6 17.4 |
| Education status<br>• No education<br>• Primary<br>• Secondary<br>• Above secondary<br>• Over all | Ref.1.5–0.1–1.6–6.1 | Ref.-8.4**-4.6***-1.5***-10.7*** | Ref.-6.94.73.1–16.8 |
| Working status<br>• Working<br>• Not working<br>• Overall | Ref.-0.7–3.9 | Ref.4.25.2 | Ref.3.51.3 |
| The ideal number of children<br>• No child<br>• 1–5<br>• 6 or more<br>• Overall | Ref.-2.40.2 | Ref.10.85.8–0.5 | Ref.8.46.00.2 |
| Concordance on the number of children<br>• Both want the same<br>• Husband want more<br>• Husband want fewer<br>• Overall | Ref.1.40.57.5 | Ref.-1.3–2.2–15.8 | Ref.0.1–1.7–8.3 |
| The desire for more children<br>• Want within two years<br>• Want after two years<br>• Other<br>• Overall | Ref.-0.3–1.2–3.7 | Ref.<br>-2.7<br>-3.7<br>-7.1 | Ref.<br>3.0<br>4.9<br>-10.8 |
| Visited by health worker<br>• No<br>• Yes<br>• Over all | Ref.-0.5–5.4 | Ref.-2.7–2.7* | Ref.3.2–8.1 |
| Total | 7.5 | 92.5 | 100.0 |

Note

*Significant at 0.05

**Significant at 0.01

***Significant at<0.001

ref: reference, LAPM: long acting and permanent methods

## Discussion

This analysis identified the presence of a significant change in LAPMs use in Ethiopia from 2005 to 2016. This finding was consistent with findings from DHS data in other sub-Saharan

African countries which showed that long-acting reversible method use increased in Malawi and Zimbabwe from 2004 to 2016 [28]. The 2015 UN estimate for trends of contraceptive use identified that LAPMs use in East Africa increased from 0.1% in 1994 to 8.6% in 2015 [4]. Increased political will, donor support, activities of non-governmental organizations, public-private partnerships and the health extension programs may all have contributed to the increase in LAPMs use in Ethiopia [12, 29].

The change in LAPMs use due to compositional factors was not significant. The reason for this may be associated with the insignificant change in the structural composition of the population of women involved in the two surveys. For example, the proportion of women who attended secondary or higher education was almost the same in the two surveys. The reports on the two EDHS also support this observation [18, 30].

Most of the changes in LAPMs use were due to the change in the effect of different characteristics of the women. The effect of age, working status, concordance between the male and female partners on the number of children, and visit by a health worker in the 12 months before the survey were significantly associated with the change in LAPMs use. About 10.0% of the increase in LAPMs use was attributable to changes in LAPMs use by women aged 35 to 49 years, a finding in agreement with other recent reports from Ethiopia [31, 32]. This might be due to a desire by older women to limit the number of children they give birth to, making them more likely to use LAPMs rather than short-acting methods. These women may have reached the planned level of fertility.

We also found that having secondary education contributed 5.0% of the observed change in LAPMs use, similar to what had been reported from other studies in Ethiopia and Kenya [13, 33–35]. Information related to LAPMs may be more accessible to women with secondary-level education, or formal education may have enabled them to develop better information processing skills, which may translate to a greater understanding of family planning messages and use of LAPMs [36]. These educated women also had an increased likelihood of being more autonomous when it came to decisions related to contraceptive use [37–39]. About 4.4% of the change in LAPMs use in this study was due to a change in the proportion of mothers who were working at the time of the survey. Working mothers may prefer using LAPMs to reduce absenteeism from work as a result of an unplanned pregnancy. Working mothers may also be better exposed to family planning messages in the workplace than those who just stay at home. Better autonomy in decision making in this group may also contribute for their increased use of reproductive health services including contraceptives.

Another important factor for the change in LAPMs use was the effect of health workers' visit. About 3% change in LAPMs use was due to the change in LAPMs use behavior among women who were visited by health workers. Studies in Ethiopia indicated that discussion about family planning with health workers was associated with increased LAPMs uptake [35, 40]. The reason for this might be that when health workers visit home, they are more likely to provide detailed information about LAPMs. The health workers may have chance to identify household level barrier for using these methods. In addition, women may be more comfortable to discuss about contraceptives at home [41]. Health care workers may have better time to counsel the mother when visiting home since health facilities are crowded in most cases. This study used the EDHS data, a nationally representative data with large sample size. The findings of the study may inform family planning programmers to strengthen home visit by health care workers to improve LAPMs uptake. The analysis did not consider program-related variables since only few of these variables were collected by the EDHS.

## Conclusion

Long-acting and permanent contraceptive methods use among married reproductive age women changed significantly between 2005 and 2016. The contribution of the variation in the survey population structure was not significant for the observed change. Most of the change in LAPMs use was associated with changes in the behavior of specific groups of women. Specifically, the change in LAPMs use was due to the behavioral change among older, educated and working women. In addition, the behavioral change among mothers who were visited by health workers contributed significantly for the change in LAPMs use. Therefore, strengthening the ongoing home visit and girls' education may have significant impact on LAPM uptake besides the other benefits these interventions had.

## Acknowledgments

We would like to acknowledge The DHS program for allowing using the EDHS data set.

## Author Contributions

**Conceptualization:** Gedefaw Abeje Fekadu, Akinyinka O. Omigbodun, Olumuyiwa A. Roberts, Alemayehu Worku Yalew.

**Data curation:** Gedefaw Abeje Fekadu, Alemayehu Worku Yalew.

**Formal analysis:** Gedefaw Abeje Fekadu, Akinyinka O. Omigbodun, Olumuyiwa A. Roberts, Alemayehu Worku Yalew.

**Investigation:** Gedefaw Abeje Fekadu, Alemayehu Worku Yalew.

**Methodology:** Gedefaw Abeje Fekadu, Olumuyiwa A. Roberts, Alemayehu Worku Yalew.

**Supervision:** Akinyinka O. Omigbodun, Olumuyiwa A. Roberts, Alemayehu Worku Yalew.

**Validation:** Gedefaw Abeje Fekadu.

**Writing – original draft:** Gedefaw Abeje Fekadu, Akinyinka O. Omigbodun, Olumuyiwa A. Roberts, Alemayehu Worku Yalew.

**Writing – review & editing:** Gedefaw Abeje Fekadu, Akinyinka O. Omigbodun, Olumuyiwa A. Roberts, Alemayehu Worku Yalew.

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
