## [Decision Letter · Decision Letter 0]

7 Oct 2019

PONE-D-19-23062

Determinants of change in long-acting and permanent contraceptive methods use in Ethiopia; a multivariate decomposition analysis of data from the Ethiopian demographic and health survey

PLOS ONE

Dear Mr. Fekadu,

Thank you for submitting your manuscript to PLOS ONE. After careful consideration, we feel that it has merit but does not fully meet PLOS ONE’s publication criteria as it currently stands. Therefore, we invite you to submit a revised version of the manuscript that addresses the points raised during the review process.

We would appreciate receiving your revised manuscript by Nov 21 2019 11:59PM. To enhance the reproducibility of your results, we recommend that if applicable you deposit your laboratory protocols in protocols.io, where a protocol can be assigned its own identifier (DOI) such that it can be cited independently in the future. For instructions see: http://journals.plos.org/plosone/s/submission-guidelines#loc-laboratory-protocols

We look forward to receiving your revised manuscript.

Kind regards,

Kannan Navaneetham

Academic Editor

PLOS ONE

Journal Requirements:

3. We noticed you not not appear to account for multiple comparisons during your statistical analysis. Please provide justification for not doing so or add these to the manuscript.

4. Please also discuss the possible limitations of your study.

Additional Editor Comments (if provided):

Reviewers' comments:

Reviewer's Responses to Questions

**Comments to the Author**

1. Is the manuscript technically sound, and do the data support the conclusions?

Reviewer #1: Yes

Reviewer #2: Yes

2. Has the statistical analysis been performed appropriately and rigorously? 

Reviewer #1: Yes

Reviewer #2: Yes

3. Have the authors made all data underlying the findings in their manuscript fully available?

Reviewer #1: Yes

Reviewer #2: Yes

4. Is the manuscript presented in an intelligible fashion and written in standard English?

Reviewer #1: No

Reviewer #2: Yes

5. Review Comments to the Author

Reviewer #1: Reviewer’s report

Title: Determinants of change in long-acting and permanent contraceptive methods use in Ethiopia; a multivariate decomposition analysis of data from the Ethiopian demographic and health survey

Version: Date: 17th September 2019

Reviewer: Samuel Bosomprah

Reviewer’s report

The paper sought to explain the change in LAPMs use over time. I believe that this is an important topic especially in developing countries where there is concern for population growth and its negative consequences on available resources and economic growth. The choice of decomposition analysis technique is appropriate to answer to the research question “why has LAMPs use change over time?”. The paper can be improved if the authors addressed the following concerns.

Major Compulsory Revisions: Yes

Introduction

Delete the first para of the introduction – it’s just a background. Begin the introduction with a clear declarative statement of the problem. The rest of the sentence in the paragraph should be an elaboration of the problem using global and sub-Saharan statistics.

Para 1 (problem statement):

Rewrite lines 57 to 71 into para 1 as follows:

“Unintended pregnancy is a major global health problem among sexually active women. <describe and="" global="" lapms="" or="" statistic="" sub-saharan="" the="" uptake="" use="" using="">. The long acting and permanent contraceptive methods (LAPMs) such as intrauterine devices or IUD, and implants are known to be efficient and cost-effective interventions for reducing unintended pregnancies. Several studies have identified <knowledge about="" lapms="">

Para 2 (justification/significance/rationale/importance): This para should describe why it is important to carry out this study. In otherwise, argue for the study. You may beef up lines 72 to 79 to reflect this thinking.

Para 3 (aims/research questions/objectives/hypotheses (if any): Conclude the introduction with the aims or objectives of the study. And indicate the potential impact of the findings from the study. Lines 80 to 88 could remain as is.

Methods

Data analysis: A number of decomposition analysis techniques exist. The authors should acknowledge same and describe which of these is implemented in the mvdcmp command. Recheck the equation in line 151.

Results

1. The tables are fragmented. The authors should collapse Tables 1 to 4 into one or two (because independent variables are many) tables. See Table 1 of the paper by Bosomprah et al 2014 (ref 23 in the current paper). In this table, the population structure was presented in the same table as the prevalence. Your last column should be maintained as the Percentage change. Include the “total” row for all tables.

2. The potential factors/drivers are too many. Authors should consider including those found in the literature to be predictors of LAPMs use or uptake. For example, I am not sure whether “sex of head of household” is a predictor of LAPMs use. Also, how is working status different from occupation? Are they not measuring the same thing? Some variables can be combined into a derived variable that makes sense. For example, can we combine: Visited by health worker in the last 12 months; Visited health facility in the last 12 months etc. I can’t see an important predictor like “knowledge about LAPMs”

3. Delete line 174 - this style is thesis report. Rather comment and put the table reference in bracket.

4. Authors should be consistent in the number of decimal places. Authors should keep the percentages in 1 decimal place in the table and the main text.

5. The comments on Table 5 (Decomposition analysis) are confusing. For example, Where are the E and C in Table 5 -as I indicated earlier Authors should include the “Total” column which should have the total E and C. Then the “coeff” are the part of the total E or C which are due to the respective predictors, also translated into percentages.

Discussion

The authors should discuss the strength and limitations of the study. They should also discuss the impact of the findings as well as prescription of future work.

Discretionary Revisions: Yes

Level of interest: An article of importance in its field

Quality of written English: Authors should proofread and correct for grammatical errors

Statistical review: No, I am a statistician and have reviewed the statistical methods used.

Declaration of competing interests: I declare that I have no competing interest

Reviewer #2: COMMENTS FOR AUTHORS/ RESEARCH TEAM

Introduction

This is a good paper and if the research team makes some changes, I think it will be worth publishing.

The authors need to consider having the manuscript edited before submission. There are several errors related to grammar and punctuation that would be caught by an editor and will also improve readability.

At some point in the introduction, the authors should provide a definition for Long acting and permanent contraceptive methods (LAPMs). They provide examples, which does help to make a distinction, but a definition will be more appropriate, I think.

A definition will also help the authors make a case for why LAPMs are superior to other forms of modern contraceptives. I assume the authors are working on the idea that LAPMs are better for efforts to lower fertility rates in Ethiopia.

The authors should decide on the preferred acronym and then stick with it. Throughout the paper, they go between LAPM and LACPM.

Methods

The authors should consider providing a definition of women who are of reproductive age (perhaps base it on the Ethiopian government's policy on national fertility). It is important to provide such a definition as that appears to be the primary inclusion criteria for data.

For example, the women's survey component of DHS data typically includes women from 15-49 years as is the case with Ethiopia. That age range falls into the reproductive age range. The authors need to confirm if this matches their notions of reproductive age ranges.

The authors chose to limit their data to married women or women cohabiting with a partner, excluding single, divorced and widowed women. I think an explanation is in order here. Contraceptive use is not something that is limited to married women and women in long-term partnerships. I can understand excluding women who are pregnant at the time of the survey, but why exclude the others? Are not the sexual health and reproductive needs of women not living with a permanent partner, pertinent to understanding the use of LAPMs?

The last paragraph of the data analysis section could be shortened a bit. For example, state from the outset that STATA was the preferred analytical software, and then note that 2005 data was appended to 2016 data to create the analysis data. Don't think there is a need to go into detail about how the data was analysis data set was created and what commands were relevant for that process.

Results

I would recommend for this manuscript the authors place the tables at the end of the document rather than incorporate them within the text. It will help improve readability I believe at this stage of the review (minor issue).

The authors use too many decimal places in their tables, particularly in what is Table 05. I can understand having several decimal places for regression coefficients, particularly if the value is small (3 decimal places should be adequate), but for percent values, these can be limited to one or two decimal places.

Consider merging Table 01 and Table 02 and within the table distinguish between the three broad categories of explanatory variables (independent variables), rather than have a separate table for demographic characteristics, a separate table for fertility-related characteristics and a separate table for family planning exposure through media. It will be a long table, but it will improve the way in which readers navigate the paper.

Similarly, the authors should consider merging Table 03 and Table 04. As suggested above, within this table the authors would distinguish between the three broad categories of independent variables.

(If the authors are feeling adventurous, it may even be possible to merge Table 01, 02, 03, and 04 into one table that is both wide and long. This table would have one major column that would be characteristics by year and then there would be two sub-columns for 2005 data and 2016 data. Similarly, there would be a larger column for Percent LAPM use per year and then sub column heading for changes in LAPM use).

Overall, the authors need to be careful about the column headings and subheadings they are using for their tables. For example, in Table 03, column headings are simply 2005 N=7918 and 2016 N=9127, when the context suggests that the authors mean, percent of LAPM use in those particular years.

Another example, in Table 05 which shows the results from the decomposition analysis, the first column is labeled as ‘LAPM contraceptive method use'. I imagine the authors simply mean variable for this particular column heading.

Lastly, does STATA not provide the totals for each variable when looking at differences in characteristics and differences in coefficients/effects? For example, there should be a total age (in years) when reporting differences in characteristics and differences in effects. There should also be an overall total for each column in Table 05. That is there should be totals reported for all differences in characteristics and differences in effects, this would correspond to what was reported in paragraph 2 of the sub-section on decomposition analysis in the results section.

Discussion

My only issue with the discussion section at present is that the authors should also consider the practical significance of their findings. Statistical significance is certainly important and worth noting and explaining. However, LAPMs in the context of a country like Ethiopia is more than whether a factor like education explains a portion of the change in use behavior. For example, the authors note that secondary education explains about 1% of the observed change in LAPM use. What does that mean for policy efforts to increase the use of LAPMs? Does it mean a focus on secondary education? But then again, though significant, a 1% change, appears to be practically small and will have little practical value for any kind of policy efforts. I am hoping that hate authors can engage these kinds of thoughts for their discussion section.

 </knowledge></describe>

6. PLOS authors have the option to publish the peer review history of their article (what does this mean?). If published, this will include your full peer review and any attached files.

Reviewer #1: Yes: Dr Samuel Bosomprah

Reviewer #2: No

---

## [Author Response · Author response to Decision Letter 0]

24 Oct 2019

We revised the manuscript based on the editor and reviewers comments. A point by point response is uploaded in the editorial manager.

---

## [Decision Letter · Decision Letter 1]

3 Dec 2019

PONE-D-19-23062R1

Determinants of change in long-acting and permanent contraceptive methods use in Ethiopia; a multivariate decomposition analysis of data from the Ethiopian demographic and health survey

PLOS ONE

Dear Mr. Fekadu,

Thank you for submitting your manuscript to PLOS ONE. After careful consideration, we feel that it has merit but does not fully meet PLOS ONE’s publication criteria as it currently stands. Therefore, we invite you to submit a revised version of the manuscript that addresses the points raised during the review process.

We would appreciate receiving your revised manuscript by Jan 17 2020 11:59PM. To enhance the reproducibility of your results, we recommend that if applicable you deposit your laboratory protocols in protocols.io, where a protocol can be assigned its own identifier (DOI) such that it can be cited independently in the future. For instructions see: http://journals.plos.org/plosone/s/submission-guidelines#loc-laboratory-protocols

We look forward to receiving your revised manuscript.

Kind regards,

Kannan Navaneetham

Academic Editor

PLOS ONE

Reviewers' comments:

Reviewer's Responses to Questions

**Comments to the Author**

1. If the authors have adequately addressed your comments raised in a previous round of review and you feel that this manuscript is now acceptable for publication, you may indicate that here to bypass the “Comments to the Author” section, enter your conflict of interest statement in the “Confidential to Editor” section, and submit your "Accept" recommendation.

Reviewer #1: All comments have been addressed

Reviewer #2: (No Response)

2. Is the manuscript technically sound, and do the data support the conclusions?

Reviewer #1: Yes

Reviewer #2: Yes

3. Has the statistical analysis been performed appropriately and rigorously? 

Reviewer #1: Yes

Reviewer #2: Yes

4. Have the authors made all data underlying the findings in their manuscript fully available?

Reviewer #1: Yes

Reviewer #2: Yes

5. Is the manuscript presented in an intelligible fashion and written in standard English?

Reviewer #1: Yes

Reviewer #2: Yes

6. Review Comments to the Author

Reviewer #1: (No Response)

Reviewer #2: This is a good paper worth publishing. The authors outline the fact that LAPM use is on the rise in Ethiopia but at less than desired rate. Furthermore, the Ethiopian public health community views the use of LAPMs as a viable, effective and safe mechanism for helping the population address the issue of unwanted pregnancies. They employ decomposition analysis, among other analytical methods, in an attempt to identify what sociodemographic factors are strongly associated with the changes in LAPM use among married/cohabiting women. Broadly, the findings point to the effect of certain factors as opposed to the structural composition of the population sampled. I believe such information is useful to those with social and public health interests on the matter of family planning and fertility.

Please see below for my comments about issues that may help to improve the quality of the manuscript further:

Introduction

I am not sure I agree with the framing in the first sentence. Are unintended pregnancies an actual public health issue? Certainly, they are a social issue and perhaps an economic one. But a public health one I am not sure about that. Perhaps there are serious public health effects that arise out of a high rate of unintended pregnancies, but I am not sure characterizing the pregnancies themselves as a health issue is wise.

What is the national target for LAPMs (Line 80)? I think it would be a good idea to state this national target if it is known or cite the policy statement that makes it a national target.

The introduction is missing a statement of aims and objectives. Traditionally, introductions will conclude with a paragraph illustrating what is intended for the rest of the manuscript. That appears to be missing in this revision. The authors get at that a bit with the last line of the paragraph (Line 84).

Methods

Personally, I believe the authors could do away with paragraph 1 in the methods or at least condense it a bit. The survey techniques employed for DHS are widely known. I think the authors could get by, by calling attention to the final reports which do a more detailed description of survey methods.

Line 100 - 106

Authors should consider providing a citation in support of the point of social desirability bias. Also, what makes the authors confident that married women or women in unions are not subject to social desirability biases when considering the issue of contraceptive use? There is an argument to be made for limiting the analysis to married/ women cohabiting with a partner, it is just not clear, at least in the text of the manuscript. The responses that the authors provided to my comment from the first review, when I raised a similar issue, could serve as a means of bolstering the argument here.

The figure legend should be placed underneath the figure. Line 107 - 109

I can appreciate the authors wanting to describe the analytical methods the chose to use. But a step-by-step descriptor of STATA commands and the way they are applied is not necessary, at least in the context of this paper. For example, where the authors write, “First the 2005 EDHS data was opened.” And then later write, “Then, the 2016 EDHS was opened.” All of that is unnecessary in my view. All those lines can be condensed to something like, “2016 EDHS data was appended to 2005 data, using the ‘append’ command”. If the authors insist on providing detail about their STATA coding choices, then perhaps consider submitting the annotated STATA do file as a supplement to be included in an appendix.

Line 166 - 167, “… and did not attend formal education.” This part of the sentence is confusing. Do the authors mean that the sampled women had no formal education or that they were not enrolled in school or some other kind of formal education at the time of the survey? I think it is the former.

In Table 01, the authors present results from the chi-square analysis. However, they make no mention of the intention to do this type of analysis in the methods section of their manuscript. Chi-square is a good choice in this case, but the authors need to explain its use in the methods.

Additionally, as a means of simplifying the table, why don’t the authors just summarize the chi-square findings in a manner similar to Table 03 (i.e., show the statistically significant findings using asterisk).

I recommend the authors take a look at Table 02. From the table, it is clear that more women are using LAPMs in 2016 than were using in 2005. However, when breaking this down by age categories the change is not as large as you do see in other categories. For 'age' we see a percent change from 2005 to 2016 of about 1.0 while for all the other variables (i.e., residence, religion, etc.) we are seeing percentage changes by categories that are very high. Why does the break down by age not account for these fairly large changes, is it an error?

Also, for Table 2 it would be a good idea to have a row at the bottom that accounted for all the women in the sample. That is the information provided in Lines 184 - 185 in the section on decomposition.

Line 226 - 227

The sentence beginning with, “About 10% of the increase…”, is confusing.

Line 228 - 229

I am not convinced by this argument, not that women in the 35-49 age bracket may want to have fewer children. But that the authors don’t put forth an explanation. Why would women in that age bracket seek fewer children on average?

Line 232 - 242

I think citing some sources may help to strengthen the authors’ arguments here. I assume that there are documented studies or interventions showing that women with an education are more likely to use LAPMs or modern contraceptives in general. For example, in line 238 where the authors suggest that LAPMs may help make Ethiopian women more economically productive, I am confident that there are economic studies demonstrating that women with a better grasp on their fertility, are more economically productive. Citing something like that gives the point the authors make in this paragraph more credibility and moves their arguments from the area of speculation.

Line 243 - 256

Similar to my comment above, I am also confident that there are interventions and studies detailed in the literature which demonstrate that some kind of home visit by some kind of medical provider/ health worker, can serve to alter behavior towards LAPMs. Citing such studies in support of the arguments here strengthens the paper in my view.

Line 253 - 256

I appreciate that the authors making this inclusion into the paper. I would suggest that the authors resist using such definitive language. Say, “The findings from this study may inform” as opposed to, “The findings from this study will inform.” There is no guarantee that policymakers will take up your work.

Did the authors remove their description of study limitations?

Suggestions for Discussion

Are there interventions that could benefit from the results of your analysis? If there are some on going on some planned, it may help to mention them in the discussion. Or the authors could suggest an intervention based on their findings, as something that merits further research.

Another potential issue of discussion is the rate of LAPMs among married men/ or men cohabiting with women. It is not the focus of your paper. However, if the argument is about informing policymaking and filling in gaps in our understanding, then it may be useful to bring up men’s use of LAPMs particularly vasectomy. The authors could bring it up in the context of future studies that need to be done in the area.

7. PLOS authors have the option to publish the peer review history of their article (what does this mean?). If published, this will include your full peer review and any attached files.

Reviewer #1: No

Reviewer #2: No

---

## [Author Response · Author response to Decision Letter 1]

6 Dec 2019

Thank you very much. A response to reviewers is uploaded

---

## [Editor Report · Decision Letter 2]

16 Dec 2019

Determinants of change in long-acting or permanent contraceptive methods use in Ethiopia; a multivariate decomposition analysis of data from the Ethiopian demographic and health survey

PONE-D-19-23062R2

Dear Dr. Fekadu,

We are pleased to inform you that your manuscript has been judged scientifically suitable for publication and will be formally accepted for publication once it complies with all outstanding technical requirements.

With kind regards,

Kannan Navaneetham

Academic Editor

PLOS ONE
---

## [Editor Report · Acceptance letter]

3 Jan 2020

PONE-D-19-23062R2 

Determinants of change in long-acting or permanent contraceptives use in Ethiopia; a multivariate decomposition analysis of data from the Ethiopian demographic and health survey 

Dear Dr. Fekadu:

I am pleased to inform you that your manuscript has been deemed suitable for publication in PLOS ONE. Congratulations! Your manuscript is now with our production department. 

With kind regards,

on behalf of

Professor Kannan Navaneetham 

Academic Editor

PLOS ONE